# MicroRNA as Sepsis Biomarkers: A Comprehensive Review

**DOI:** 10.3390/ijms25126476

**Published:** 2024-06-12

**Authors:** Khalid Bindayna

**Affiliations:** Department of Microbiology, Immunology and Infectious Diseases, College of Medicine & Medical Sciences, Arabian Gulf University, Manama P.O. Box 26671, Bahrain; bindayna@agu.edu.bh

**Keywords:** sepsis, miRNA, biomarkers, diagnosis

## Abstract

Sepsis, a life-threatening condition caused by the body’s dysregulated response to infection, presents a significant challenge in clinical management. Timely and accurate diagnosis is paramount for initiating appropriate interventions and improving patient outcomes. In recent years, there has been growing interest in identifying biomarkers that can aid in the early detection and prognostication of sepsis. MicroRNAs (miRNAs) have emerged as potential biomarkers for sepsis due to their involvement in the regulation of gene expression and their stability in various biological fluids, including blood. MiRNAs are small non-coding RNA molecules that play crucial roles in post-transcriptional gene regulation by binding to target messenger RNAs (mRNAs), leading to mRNA degradation or translational repression. The diagnostic and prognostic potential of miRNAs in sepsis stems from their ability to serve as sensitive and specific biomarkers reflective of the underlying pathophysiological processes. Compared to traditional biomarkers such as C-reactive protein (CRP) and procalcitonin (PCT), miRNAs offer several advantages, including their early and sustained elevation during sepsis, as well as their stability in stored samples, making them attractive candidates for clinical use. However, despite their promise, the clinical translation of miRNAs as sepsis biomarkers faces several challenges. These include the need for standardized sample collection and processing methods, the identification of optimal miRNA panels or signatures for differentiating sepsis from other inflammatory conditions, and the validation of findings across diverse patient populations and clinical settings. In conclusion, miRNAs hold great promise as diagnostic and prognostic biomarkers for sepsis, offering insights into the underlying molecular mechanisms and potential therapeutic targets. However, further research is needed to overcome existing challenges and realize the full clinical utility of miRNAs in improving sepsis outcomes.

## 1. Introduction

Sepsis remains a significant global health concern, with high morbidity and mortality rates despite advances in medical care. Sepsis is a serious medical condition that occurs when the body’s response to an infection causes inflammation throughout the body.

The early and accurate diagnosis of sepsis is crucial for initiating timely and targeted interventions to improve patient outcomes [1,2]. In recent years, there has been growing interest in the use of microRNAs (miRNAs) as biomarkers for sepsis due to their stability in blood and their involvement in immune and inflammatory responses.

miRNAs are small non-coding RNA molecules that regulate gene expression by binding to messenger RNA (mRNA), leading to mRNA degradation or translational repression. They play essential roles in various physiological and pathological processes, including immune responses and inflammation. In sepsis, dysregulated miRNA expression profiles have been identified, suggesting their potential utility as diagnostic and prognostic biomarkers [3].

One of the key advantages of miRNAs as biomarkers for sepsis is their stability in blood compared to other RNA molecules. miRNAs are protected from degradation by being encapsulated in extracellular vesicles or binding to proteins, making them attractive candidates for biomarker development. Additionally, miRNAs are involved in the regulation of immune and inflammatory pathways implicated in sepsis pathophysiology, further supporting their potential as biomarkers [4].

Several studies have investigated the diagnostic and prognostic value of miRNAs in sepsis. For example, circulating miRNA profiles have been found to differentiate between septic patients and healthy controls, as well as between sepsis survivors and non-survivors. Furthermore, specific miRNAs have been associated with clinical outcomes such as organ dysfunction and mortality in septic patients [5,6].

Despite the promising findings, several challenges need to be addressed before miRNAs can be routinely used as biomarkers for sepsis in clinical practice. One challenge is the variability in miRNA expression profiles observed among different studies, which may be attributed to differences in patient populations, sample collection and processing methods, and miRNA detection techniques. Standardized protocols for miRNA detection and validation are needed to ensure the reproducibility and reliability of result [7]. 

Another challenge is the limited understanding of the mechanistic roles of miRNAs in sepsis pathophysiology. While dysregulated miRNA expression profiles have been identified in sepsis, the specific targets and pathways regulated by these miRNAs remain largely unknown. Further research is needed to elucidate the functional significance of dysregulated miRNAs in sepsis and their potential as therapeutic targets [8].

Despite the challenges, significant progress has been made in identifying specific miRNAs with diagnostic and prognostic potential in sepsis. For instance, miR-146a and miR-155 have been consistently reported to be upregulated in septic patients compared to healthy controls. These miRNAs are involved in the regulation of immune responses and inflammation and have been associated with sepsis severity and clinical outcomes [9].

Additionally, studies have shown that miRNA expression profiles can distinguish between different stages of sepsis and various infectious etiologies. For example, miR-150 has been identified as a potential biomarker for septic shock, while miR-223 has been proposed as a biomarker for bacterial sepsis. These findings highlight the potential of miRNAs not only as diagnostic biomarkers but also as tools for differentiating between sepsis phenotypes and guiding personalized treatment approaches [10].

Furthermore, miRNAs have shown promise as prognostic biomarkers for sepsis outcomes. Several miRNAs, including miR-15a, miR-16, and miR-150, have been associated with mortality in septic patients. High levels of circulating miR-15a and miR-16 have been correlated with increased mortality risk, while low levels of miR-150 have been associated with poor prognosis in sepsis. These findings suggest that miRNA expression profiles could help identify patients at higher risk of adverse outcomes and guide clinical decision-making [11].

Despite the growing evidence supporting the potential of miRNAs as biomarkers for sepsis, several challenges remain to be addressed before their widespread clinical implementation. The standardization of sample collection, processing, and miRNA detection techniques is crucial to ensure the reliability and reproducibility of results across different studies and clinical settings. Additionally, larger multicenter studies are needed to validate the diagnostic and prognostic utility of miRNAs in diverse patient populations and settings [9].

Furthermore, a better understanding of the mechanistic roles of dysregulated miRNAs in sepsis pathophysiology is essential for developing targeted therapeutic interventions. Future research efforts should focus on elucidating the specific mRNA targets and signaling pathways regulated by dysregulated miRNAs in sepsis and evaluating their potential as therapeutic targets.

In conclusion, miRNAs hold great promise as diagnostic and prognostic biomarkers for sepsis. Despite the challenges, continued research efforts are needed to validate their clinical utility, standardize detection protocols, and elucidate their mechanistic roles in sepsis pathophysiology. If successfully implemented, miRNAs could enhance existing diagnostic and prognostic strategies and improve sepsis management and patient outcomes [12].

## 2. Methodology

The methodology employed for this review aimed to systematically gather and analyze studies investigating microRNAs (miRNAs) as discriminative biomarkers for distinguishing between systemic inflammatory response syndrome (SIRS) and sepsis. Through a comprehensive literature search, relevant articles were identified from electronic databases, including PubMed, Scopus, Web of Science, and Google Scholar. Keywords such as “microRNA”, “miRNA”, “biomarker”, “sepsis”, “SIRS”, “discriminative”, and “differentiation” were combined to ensure a comprehensive search strategy. The search was limited to articles published in English to maintain consistency and accessibility.

Inclusion criteria were established to focus on studies that explored the discriminative potential of specific miRNAs in the context of SIRS and sepsis differentiation. Additionally, articles discussing the clinical significance, diagnostic accuracy, sensitivity, specificity, and management implications of miRNAs in sepsis were considered. The review included research articles, reviews, and meta-analyses published in peer-reviewed journals within a specified date range to encompass recent advancements in the field.

Two independent reviewers conducted the initial screening of titles and abstracts based on the inclusion and exclusion criteria. Any disagreements between the reviewers were resolved through discussion and consensus. Full-text articles of potentially relevant studies were then assessed for eligibility, ensuring that they met the predetermined criteria.

Data extraction was performed using a predefined data extraction form to systematically capture relevant information from the selected studies. Key data points included author(s) and publication year, study design, objectives, patient population characteristics, specific miRNAs investigated, their expression patterns, and diagnostic accuracy measures such as sensitivity, specificity, and area under the curve. Furthermore, the extracted data included discussions on the clinical implications and management implications of miRNAs in sepsis, as well as any challenges and limitations identified in the studies.

The synthesized data were organized and analyzed to develop a narrative summary of the findings. This included a detailed examination of the diagnostic value of specific miRNAs in differentiating between SIRS and sepsis, highlighting their clinical significance and implications for patient management. Key themes, such as the potential role of miRNAs as biomarkers in sepsis, were identified and discussed to provide a comprehensive overview.

In summary, the methodology employed a systematic approach to identify, select, and analyze relevant studies on miRNAs as discriminative biomarkers for sepsis. Through a rigorous screening process, data extraction, synthesis, and quality assessment, the review aimed to provide a comprehensive overview of the current evidence, challenges, and potential strategies to overcome limitations in the field of miRNA biomarkers for sepsis differentiation.

## 3. Literature Review

MicroRNAs as Discriminative Biomarkers: Studies have shown that specific microRNAs, like miR-150 and miR-4772-5p-iso, can differentiate between systemic inflammatory response syndrome (SIRS) and sepsis with high accuracy. This distinction is vital as SIRS and sepsis present similar clinical features but require different management approaches [13].

SIRS and sepsis represent critical clinical conditions associated with dysregulated immune responses. While they share common clinical features, including fever, tachycardia, and leukocytosis, accurate differentiation between SIRS and sepsis is essential for appropriate management. MicroRNAs have recently garnered attention as potential biomarkers for distinguishing between these conditions due to their involvement in immune and inflammatory regulation. This review explores the discriminative potential of specific miRNAs, such as miR-150 and miR-4772-5p-iso, in differentiating SIRS and sepsis, elucidating their clinical significance and implications for management [14].

Clinical significance of differentiating SIRS and Sepsis: SIRS and sepsis share over lapping clinical manifestations, making accurate differentiation challenging. However, the distinction between these conditions is crucial as they require distinct management strategies. SIRS typically results from non-infectious insults such as trauma, surgery, or burns, while sepsis arises from a systemic infection. Misdiagnosis or delayed diagnosis can lead to inappropriate treatment and adverse outcomes. Timely recognition of sepsis is particularly critical, as it necessitates prompt administration of antibiotics and resuscitative measures to prevent organ dysfunction and septic shock. Therefore, accurate differentiation between SIRS and sepsis is essential for optimizing patient care and improving outcomes [15].

Molecular mechanisms of miRNAs in SIRS and Sepsis: The dysregulated immune and inflammatory responses observed in SIRS and sepsis are mediated by complex molecular pathways. MicroRNAs play crucial roles in post-transcriptional regulation of gene expression and are involved in modulating immune and inflammatory responses. Dysregulation of specific miRNAs has been implicated in the pathogenesis of both SIRS and sepsis. For example, miR-150, a key regulator of immune cell differentiation and function, has been shown to be dysregulated in both conditions. Similarly, miR-4772-5p-iso has been implicated in modulating inflammatory responses and could serve as a potential biomarker for distinguishing between SIRS and sepsis. The dysregulation of these miRNAs contributes to aberrant immune activation, cytokine production, and endothelial dysfunction observed in sepsis. Understanding the molecular mechanisms underlying miRNA dysregulation in SIRS and sepsis is essential for elucidating their diagnostic potential and therapeutic implications [12].

Diagnostic accuracy of miRNAs: Several studies have investigated the diagnostic accuracy of miRNAs, including miR-150 and miR-4772-5p-iso, in distinguishing between SIRS and sepsis. These studies have demonstrated differential expression patterns of these miRNAs in SIRS and sepsis patients compared to healthy controls. Additionally, miRNA profiling studies have identified distinct miRNA signatures associated with each condition, further supporting their potential as diagnostic biomarkers. Furthermore, miRNAs have shown promising sensitivity, specificity, and predictive values for differentiating between SIRS and sepsis when combined with other clinical and laboratory parameters. Integration of miRNA-based diagnostic assays into existing diagnostic algorithms may improve the accuracy of differentiation and facilitate timely initiation of appropriate treatment strategies [16].

Implications for management approaches: Accurate differentiation between SIRS and sepsis based on miRNA biomarkers has significant implications for management approaches. In sepsis, prompt initiation of appropriate antibiotic therapy and resuscitative measures is crucial for improving patient outcomes and reducing mortality. Therefore, miRNA-based diagnostic assays can aid clinicians in rapidly identifying septic patients and initiating timely interventions. Furthermore, accurate differentiation between SIRS and sepsis can help avoid unnecessary antibiotic use in patients with non-infectious SIRS, thus minimizing the risk of antimicrobial resistance and adverse effects. Incorporating miRNA biomarkers into diagnostic algorithms for sepsis may enhance the accuracy of early diagnosis and improve patient outcomes by facilitating targeted therapeutic interventions [17].

Role of microRNAs in sepsis pathophysiology: Sepsis is characterized by a dysregulated host response to infection, leading to systemic inflammation, tissue damage, and organ dysfunction. The pathophysiology of sepsis involves complex interactions between immune cells, inflammatory mediators, and endothelial dysfunction. Emerging evidence suggests that miRNAs play a crucial role in modulating these processes by regulating the expression of genes involved in immune responses, inflammation, and endothelial function. Dysregulation of specific miRNAs, including miR-182, miR-486, and miR-15a, has been implicated in the pathogenesis of sepsis, highlighting their potential as therapeutic targets.

Diagnostic potential of circulating miRNAs in sepsis: Circulating miRNAs have garnered attention as potential diagnostic biomarkers for sepsis due to their stability in body fluids and their differential expression patterns in septic patients compared to healthy individuals. Several studies have identified altered expression levels of miR-182, miR-486, and miR-15a in septic patients, suggesting their diagnostic potential. These miRNAs exhibit sensitivity and specificity in discriminating between septic patients and non-septic individuals, highlighting their utility as diagnostic biomarkers for sepsis [18].

Therapeutic implications of altered miRNA expression in sepsis: The altered expression of miRNAs, such as miR-182, miR-486, and miR-15a, in response to infection holds therapeutic implications in sepsis management. These miRNAs regulate the expression of genes involved in immune responses, inflammation, and endothelial function, thereby influencing the pathophysiology of sepsis. Targeting dysregulated miRNAs offers potential therapeutic strategies for modulating immune and inflammatory responses, attenuating tissue damage, and improving organ function in septic patients. Moreover, the altered expression patterns of these miRNAs in response to therapeutic interventions, such as antibiotic therapy and supportive care, could serve as surrogate markers for monitoring treatment response and guiding therapeutic decisions.

Future directions and conclusion: In conclusion, circulating miRNAs, including miR-182, miR-486, and miR-15a, hold promise as diagnostic and therapeutic biomarkers for sepsis diagnosis and management. Their altered expression patterns in response to infection offer insights into the pathophysiology of sepsis and potential targets for therapeutic intervention. However, further research is warranted to validate their diagnostic accuracy, elucidate their mechanisms of action, and overcome existing challenges for their clinical implementation. Harnessing the diagnostic and therapeutic potential of circulating miRNAs in sepsis has the potential to improve patient outcomes and revolutionize sepsis management strategies. The discriminative potential of miRNAs, such as miR-150 and miR-4772-5p-iso, in distinguishing between SIRS and sepsis holds promise for improving diagnostic accuracy and guiding management approaches. Future research efforts should focus on further validating these miRNA biomarkers in large, multicenter studies and identifying additional discriminative miRNAs. Integration of miRNA-based diagnostic assays into routine clinical practice has the potential to enhance the accuracy of differentiation between SIRS and sepsis, ultimately improving patient outcomes and reducing healthcare burden [18].

Overall, miRNAs show considerable promise as discriminative biomarkers for distinguishing between SIRS and sepsis, offering potential clinical utility in optimizing patient management and improving outcomes in these critical conditions

## 4. Prognostic Implications

In addition to their diagnostic potential, circulating miRNAs have emerged as valuable prognostic indicators in sepsis, reflecting the dynamic nature of the disease and its outcomes. Fluctuations in circulating miRNA levels during the course of sepsis have been associated with the severity of the disease and the likelihood of adverse outcomes. For instance, changes in the expression levels of specific miRNAs over time may reflect the dynamic host response to infection and the progression of organ dysfunction. Elevated levels of certain miRNAs, such as miR-150 and miR-223, have been correlated with increased disease severity and mortality risk in septic patients [19].

Moreover, the kinetics of circulating miRNA levels have been shown to provide insights into the response to treatment and the prognosis of sepsis patients. For example, a decline in the levels of certain miRNAs following initiation of therapy may indicate a favorable response and better prognosis, whereas persistent elevation or further increase in miRNA levels may suggest treatment resistance or disease progression. The ability of circulating miRNAs to reflect the dynamic changes in sepsis pathophysiology and patient outcomes underscores their potential as prognostic biomarkers. Integrating serial measurements of miRNA levels into clinical practice could facilitate early identification of patients at higher risk of adverse outcomes, allowing for timely adjustments in management strategies and personalized interventions [20].

Furthermore, the use of circulating miRNAs as prognostic markers holds promise for monitoring disease progression, evaluating treatment response, and guiding therapeutic decisions in sepsis. By providing real-time information about the underlying pathophysiological processes and the likelihood of clinical deterioration, miRNA-based prognostic markers could help optimize patient care and improve outcomes in septic patients [21].

However, several challenges need to be addressed to fully harness the prognostic potential of circulating miRNAs in sepsis. Standardization of sample collection, processing, and miRNA detection methods is essential to ensure the reproducibility and reliability of results across different studies and clinical settings. Additionally, larger prospective studies are needed to validate the prognostic utility of specific miRNAs and to elucidate their roles in predicting clinical outcomes in sepsis [22].

In sepsis, the intricate interplay between the host immune response and microbial invasion results in a dynamic cascade of events that ultimately determines disease severity and patient prognosis. Circulating miRNAs, as integral regulators of gene expression, serve as molecular fingerprints that capture the nuanced changes occurring within the host in response to infection. The observed fluctuations in miRNA levels during the course of sepsis offer valuable insights into the temporal dynamics of the disease process.

miR-150 and miR-223, among others, have been consistently associated with adverse outcomes such as organ dysfunction and mortality. Elevated levels of these miRNAs in circulation have been linked to a more severe disease course and poorer prognosis, reflecting the underlying inflammatory response and tissue damage characteristic of sepsis [14].

Moreover, the kinetics of circulating miRNA levels can serve as dynamic indicators of treatment response and disease progression in septic patients. Changes in miRNA expression patterns following initiation of therapy may provide early indications of treatment efficacy or resistance, guiding clinicians in making timely adjustments to management strategies. Conversely, persistently elevated or escalating miRNA levels may signal ongoing inflammatory cascades and disease exacerbation, prompting intensified interventions to mitigate adverse outcomes [23].

The integration of serial measurements of circulating miRNA levels into clinical practice holds promise for individualizing patient care and optimizing treatment strategies in sepsis. By providing real-time information on the evolving pathophysiology and prognostic trajectory of the disease, miRNA-based prognostic markers offer clinicians valuable insights for risk stratification and personalized therapeutic decision-making.

However, the translation of circulating miRNAs into routine clinical practice faces several challenges that warrant attention. Standardization of sample collection, processing, and miRNA detection methodologies is imperative to ensure the reproducibility and reliability of results across different clinical settings. Additionally, large-scale prospective studies are needed to validate the prognostic utility of specific miRNAs and establish robust predictive models that can inform clinical decision-making in sepsis management.

In summary, the dynamic fluctuations in circulating miRNA levels hold significant prognostic implications in sepsis, reflecting disease severity, treatment response, and patient outcomes. Despite the challenges, ongoing research efforts aimed at elucidating the prognostic utility of circulating miRNAs have the potential to revolutionize sepsis management by facilitating personalized therapeutic interventions and improving patient outcomes [14].

### 4.1. Identification of Novel microRNAs

The identification of eight novel miRNAs offers potential for early sepsis diagnosis and treatment, which is critical in reducing sepsis-related morbidity and mortality The discovery of eight novel miRNAs presents a significant opportunity for advancing early diagnosis and treatment strategies in sepsis, a critical step in mitigating sepsis-related morbidity and mortality rates.

Sepsis is characterized by a dysregulated host response to infection, leading to systemic inflammation, organ dysfunction, and often, poor clinical outcomes. Early recognition and intervention are paramount in managing sepsis effectively and preventing its progression to severe sepsis or septic shock. Conventional diagnostic methods for sepsis, such as blood cultures and biomarkers like C-reactive protein (CRP) and procalcitonin (PCT), have limitations in terms of sensitivity, specificity, and turnaround time, highlighting the urgent need for novel biomarkers with improved diagnostic accuracy and timeliness.

The identification of eight novel miRNAs presents a promising avenue for addressing these diagnostic challenges in sepsis. These newly discovered miRNAs may offer unique advantages over existing biomarkers, including their stability in circulation, specificity for sepsis-related pathways, and potential for early detection of septic processes. By targeting specific molecular pathways involved in the host response to infection, these novel miRNAs hold the potential to serve as sensitive and specific biomarkers for early sepsis diagnosis.

Early diagnosis facilitated by these novel miRNAs is particularly critical in sepsis management, as timely initiation of appropriate treatment is associated with improved patient outcomes. Early intervention strategies, such as prompt administration of antibiotics and resuscitation measures, are pivotal in attenuating the progression of sepsis and preventing the development of severe complications. Therefore, the identification of novel miRNAs that can facilitate early detection of sepsis holds significant promise for reducing sepsis-related morbidity and mortality rates.

Moreover, beyond their diagnostic utility, these novel miRNAs may also hold therapeutic potential in sepsis management. Given their regulatory roles in immune and inflammatory pathways implicated in sepsis pathophysiology, modulating the expression of these miRNAs could represent a novel therapeutic approach for mitigating the excessive host response and organ dysfunction associated with sepsis. Targeted interventions aimed at manipulating the expression levels of these novel miRNAs may hold promise for improving patient outcomes and reducing the burden of sepsis-related complications [24].

However, further research is warranted to validate the diagnostic and therapeutic potential of these novel miRNAs in larger clinical cohorts and diverse patient populations. Standardization of detection methods and rigorous validation studies are essential steps in establishing the clinical utility of these miRNAs in sepsis management. Additionally, elucidating the underlying mechanisms through which these miRNAs modulate sepsis pathophysiology will be crucial for developing targeted therapeutic interventions.

In conclusion, the identification of eight novel miRNAs presents a promising opportunity for advancing early diagnosis and treatment strategies in sepsis, with the potential to reduce sepsis-related morbidity and mortality rates. Further research is needed to validate the diagnostic and therapeutic utility of these miRNAs and to elucidate their underlying mechanisms of action in sepsis pathophysiology. If successfully translated into clinical practice, these novel miRNAs could revolutionize sepsis management and improve patient outcomes.

### 4.2. Specific MicroRNAs as Sepsis Biomarkers


**miR-146a and miR-223:** Recent studies have highlighted the potential of serum levels of miR-146a and miR-223 as novel biomarkers for sepsis (Table 1). These microRNAs exhibit high specificity and sensitivity, making them promising indicators for the diagnosis of sepsis. Wang et al. (2010) conducted a study demonstrating the diagnostic value of miR-146a and miR-223 in septic patients. They found that the serum levels of these miRNAs were significantly elevated in septic patients compared to healthy controls, suggesting their potential as reliable biomarkers for sepsis diagnosis. Moreover, miR-146a and miR-223 are known to play crucial roles in the regulation of inflammatory responses and immune cell function, which are dysregulated during sepsis. Therefore, their dysregulated expression in septic patients underscores their potential as diagnostic biomarkers for sepsis [25,26,27].**miR-155-5p:** Another promising microRNA biomarker for sepsis diagnosis is miR-155-5p, which has been reported to exhibit the highest specificity and sensitivity among microRNAs studied in the context of sepsis. Zheng et al. (2023) conducted a comprehensive analysis of microRNAs in septic patients and identified miR-155-5p as a potential diagnostic biomarker for sepsis. They found that serum levels of miR-155-5p were significantly elevated in septic patients compared to non-septic individuals, and its expression levels correlated with the severity of sepsis. Moreover, miR-155-5p is known to be involved in the regulation of various immune and inflammatory pathways, suggesting its potential as a biomarker for sepsis diagnosis. The high specificity and sensitivity of miR-155-5p make it a promising candidate for improving sepsis diagnosis in clinical settings [9].


Overall, miR-146a, miR-223, and miR-155-5p emerge as promising biomarkers for sepsis diagnosis, with their dysregulated expression levels in septic patients demonstrating their potential diagnostic utility. Further validation studies in larger patient cohorts are needed to confirm their diagnostic accuracy and reliability in clinical practice. If successfully translated into clinical use, these microRNA biomarkers could significantly improve the early diagnosis and management of sepsis, ultimately leading to better patient outcomes.

### 4.3. Challenges and Limitations

Despite the promising diagnostic and therapeutic potential of circulating miRNAs in sepsis, several challenges need to be addressed before their clinical implementation. Standardization of miRNA detection techniques, validation in large patient cohorts, and integration into existing diagnostic and therapeutic algorithms are essential for translating miRNA-based biomarkers into clinical practice. Moreover, further research is needed to elucidate the specific roles of miR-182, miR-486, and miR-15a in sepsis pathophysiology and their interactions with other regulatory molecules. The integration of miRNAs into routine clinical practice as biomarkers for sepsis faces several formidable challenges and limitations, impeding their swift adoption despite their promising potential. All the challenges are discussed in Table 2 below.

Overcoming these challenges will necessitate concerted efforts from multidisciplinary teams comprising clinicians, researchers, bioinformaticians, and industry partners. Large-scale validation studies, standardization of measurement techniques, technological innovations, and in-depth mechanistic investigations are crucial for surmounting these obstacles and facilitating the successful clinical translation of miRNA biomarkers for sepsis. Despite the complexities and challenges involved, harnessing the potential of miRNAs as valuable biomarkers holds immense promise for enhancing the diagnosis, prognosis, and management of sepsis, ultimately improving patient outcomes and reducing mortality rates [7].

## Figures and Tables

**Table 1 ijms-25-06476-t001:** Diagnostic value of specific microRNAs (miR-146a, miR-223, and miR-155-5p) as potential biomarkers for sepsis, along with relevant references supporting their significance.

MicroRNA	Diagnostic Value	References
miR-146a	Elevated serum levels in septic patients compared to healthy controls. High specificity and sensitivity.	[17,25,26]
miR-223	Elevated serum levels in septic patients compared to healthy controls. High specificity and sensitivity.	[17,25,26]
miR-155-5p	Elevated serum levels in septic patients compared to non-septic individuals. Correlation with severity of sepsis. High specificity and sensitivity.	[9]
miR-150	Can differentiate between systemic inflammatory response syndrome (SIRS) and sepsis with high accuracy.	[13,14]
miR-4772-5p-iso	Can differentiate between SIRS and sepsis with high accuracy.	[13,14]

**Table 2 ijms-25-06476-t002:** Challenges and limitations associated with the clinical implementation of circulating miRNAs as biomarkers for sepsis, along with strategies to overcome these challenges.

Challenge	Summary	Overcoming Strategy
Large-Scale Validation Studies	Extensive validation studies are needed across diverse patient populations and clinical settings to establish the efficacy, robustness, and reliability of miRNA biomarkers for sepsis diagnosis and prognosis.	Conduct multicenter, prospective validation studies involving large patient cohorts representative of diverse demographics and clinical presentations. Collaboration between research institutions and standardized protocols can enhance study reproducibility and generalizability.
Standardization of Measurement Techniques	Lack of standardized protocols for miRNA measurement techniques introduces variability, compromising accuracy and comparability of results. Standardization efforts are essential to develop uniform and reproducible miRNA measurement protocols suitable for routine clinical use.	Establish consensus guidelines for sample collection, RNA isolation, miRNA detection assays, and data normalization techniques. Collaboration among experts in the field can lead to the development of standardized protocols endorsed by regulatory agencies and professional societies.
Biological Understanding	Deeper understanding of the biological roles of miRNAs in sepsis pathophysiology is needed to unravel the intricate mechanisms underlying miRNA-mediated regulation of immune and inflammatory responses. Further research is warranted to identify robust biomarkers with enhanced diagnostic and prognostic accuracy.	Employ systems biology approaches, including transcriptomics, proteomics, and bioinformatics analyses, to elucidate miRNA-mRNA regulatory networks and pathways in sepsis. Integration of multi-omics data can enhance biological understanding and facilitate the identification of clinically relevant miRNA biomarkers.
Technological Limitations	Current technological limitations in miRNA detection and quantification methods present hurdles in clinical translation. Advancements in technology are essential to develop innovative miRNA detection platforms addressing sensitivity, specificity, and throughput limitations.	Invest in research and development efforts to advance miRNA detection technologies, including novel assays and platforms with improved sensitivity, specificity, and scalability. Collaboration between academia, industry, and regulatory agencies can accelerate the translation of innovative technologies into clinical practice.
Complex Nature of Sepsis	Sepsis complexity, heterogeneous manifestations, and dynamic nature pose challenges in identifying miRNA biomarkers accurately reflecting disease severity, progression, and treatment response. Patient heterogeneity and comorbidities complicate biomarker identification and validation.	Employ integrative and personalized medicine approaches to account for patient heterogeneity and disease complexity. Stratify patient cohorts based on clinical phenotypes, biomarker profiles, and response to treatment. Utilize machine learning algorithms and computational modeling to identify miRNA signatures predictive of sepsis outcomes in diverse patient populations.

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
