# Peer review of "MicroRNA as Sepsis Biomarkers: A Comprehensive Review"

_ijms, 2024, doi:10.3390/ijms25126476_

Round 1

Reviewer 1 Report

Comments and Suggestions for Authors

The authors aim to create a summary of the current literature on the topic of miRNA as biomarkers in sepsis. Considering the numerous literature and studies available on this topic, this is a good idea that could certainly benefit the research field. However, the authors unfortunately fail to summarize truly new information in this review. The content and the all in all statements/take-home messages are similar to those of other reviews/ meta-analysis already published on this topic (Benz eta al. 2016; Int. J. Mol. Sci. 2016; Mi-Hee-Kim et al., Infect Chemother. 2020 Mar;52(1):1-18; Shen et al. Journal of Intensive Care (2020) 8:84; Zheng X et al., 2023, PLoS ONE 18(2): e0279726) and thus do not currently provide any new input for this scientific field.

The authors start with a clear introduction (paragraph 1). Subsequently, several sections follow (chapter 2 - 3 with “sub-sections”). But unfortunately, this sections repeatedly reiterate contents from the introduction (paragraph 1) or previous chapters (e.g. line 121-122 is the same like line 107-113) and rarely provide anything new. All these sections (paragraph 1 – 3.1.) could be summarized as an introduction, followed by a discussion of various miRNAs (like the section 3.2.). However, it would be appropriate here to mention truly new studies with new miRNAs (e.g. Baobin Sun et.al. Journal of Inflammation research, 2021:14 3687–3695) The ones listed here have already been mentioned in other reviews (Benz eta al. 2016; Int. J. Mol. Sci. 2016, 17, 78; doi:10.3390/ijms17010078), and there is no new information contained in this chapter.

A section on the problems associated with using miRNA as biomarkers and potential solutions could be a good idea as the last chapter in this review (like paragraph 3.3.). However, there are also too many repetitions from the preceding chapters, and references are missing in some places. The authors should also decide whether they want to present it in tabular form or as continuous text.

Conclusion: The review needs to be completely revised, addressing:

Avoiding content repetitions.

Providing missing references (e.g. line 179, sometimes they write “…several studies…” like in lien 246 but now references behind the sentence or only one reference at the end of the section)

Summarizing genuinely new studies (new miRNAs...).

Adding a methodology section that includes details on literature search.

Author Response

Thank you for reviewing my review paper.

I tried to work on the repetitions. 

Added the missing references.

Added methodology section.

Some repetitions can't be deleted as these miRNAs work as both diagnostic and prognostic markers.

Reviewer 2 Report

Comments and Suggestions for Authors

Bindayna carried out a literature review on sepsis biomarkers. The review is comprehensive and helpful to the reader. I only have a few suggestions.

a brief definition of sepsis should be included in the introduction

several problems and challenges are reported in the introduction which should instead be reported at the end of the text. The introduction should be concise and highlight the need for a literature review

The methods section is missing

The text ends with challenges, but a real conclusion is missing. 

Author Response

Thank you for reviewing my review paper.

Added definition of sepsis. 

Challenges are added and last also with possible strategies for overcoming those.

Mehodology part added.